# Designing temporal networks that synchronize under resource constraints

Yuanzhao Zhang [1✉] & Steven H. Strogatz [1✉]

Being fundamentally a non-equilibrium process, synchronization comes with unavoidable energy costs and has to be maintained under the constraint of limited resources. Such resource constraints are often reflected as a finite coupling budget available in a network to facilitate interaction and communication. Here, we show that introducing temporal variation in the network structure can lead to efficient synchronization even when stable synchrony is impossible in any static network under the given budget, thereby demonstrating a fundamental advantage of temporal networks. The temporal networks generated by our open-loop design are versatile in the sense of promoting synchronization for systems with vastly different dynamics, including periodic and chaotic dynamics in both discrete-time and continuous-time models. Furthermore, we link the dynamic stabilization effect of the changing topology to the curvature of the master stability function, which provides analytical insights into synchronization on temporal networks in general. In particular, our results shed light on the effect of network switching rate and explain why certain temporal networks synchronize only for intermediate switching rate.

[1] Center for Applied Mathematics, Cornell University, Ithaca 14853 NY, USA. ✉email: yuanzhao@u.northwestern.edu; strogatz@cornell.edu

Synchronization is critical to the function of many interconnected systems[1], from physical[2] to technological[3] and biological[4]. Many such systems need to synchronize under the constraint of limited resources. For instance, energy dissipation is required to couple molecular biochemical oscillators through oscillator–oscillator exchange reactions, which are responsible for synchronization in systems such as the cyanobacterial circadian clock[5]. For multiagent networks with distributed control protocols, including robotic swarms, the synchronization performance is limited by the available budget of control energy[6].

Similarly, for networks of coupled oscillators, one important resource is the total coupling budget[7], which determines how strongly the oscillators can influence each other. For a typical oscillator network, a minimum coupling strength $\sigma_c$ is needed to overcome transversal instability and maintain synchronization. The network structures that achieve synchronization with the minimum coupling strength are optimal, and they are characterized by a complete degenerate spectrum[8]—all eigenvalues of the Laplacian matrix are identical, except the trivial zero eigenvalue associated with perturbations along the synchronization trajectory. Below $\sigma_c$, there is no network structure that can maintain synchrony without violating the resource constraint.

The results above, however, are derived assuming the network to be static. That is, the network connections do not change over time. Previous studies have shown that temporal networks[9–15] can synchronize better than two of their static counterparts—namely, those obtained either by freezing the network at given time instants[16–19] or by averaging the network structure over time[20–22]. But it remains unclear whether there are temporal networks that can outperform all possible static networks. In particular, can temporal variations synchronize systems beyond the fundamental limit set by the optimal static networks? This question is especially interesting given that past studies have often focused on the fast-switching limit, for which the network structure changes much faster than the node dynamics. These fast-switching networks are equivalent to their static, time-averaged counterparts in terms of synchronization stability[17,23–25]. Thus, no temporal networks can outperform optimal static networks in the fast-switching limit.

In this article, we show that the full potential of temporal networks lies beyond the fast-switching limit, a message echoed by several recent studies[21,26,27]. Importantly, by allowing a network to vary in time at a suitable rate, synchronization can be maintained even when the coupling strength is below $\sigma_c$ for all time $t$. We also develop a general theory to characterize the synchronizability of commutative temporal networks. The use of commutative graphs in synchronization was pioneered in refs. [18,20] and subsequently adopted in numerous studies[22,26,28] for its potential of generating analytical insights beyond the fast-switching limit. An insight provided by our theory is that the effectiveness of introducing time-varying coupling depends critically on the curvature of the master stability function[29] at its first zero, which extends the results presented in ref. [30]. Moreover, we demonstrate analytically that the condition for improved synchronizability in temporal networks is universally satisfied by coupled one-dimensional maps.

## Results

### Networks of coupled oscillators

We start by considering systems described by the following dynamical equations:

$$\dot{\mathbf{x}}_i = \mathbf{F}(\mathbf{x}_i) - \sigma \sum_{j=1}^{n} L_{ij}(t)\mathbf{H}(\mathbf{x}_j), \quad i = 1, \dots, n, \tag{1}$$

where $\mathbf{L} = (L_{ij})$ is the normalized Laplacian matrix representing a diffusively coupled network. Here, $L_{ij} = \delta_{ij}\sum_k A_{ik} - A_{ij}$, with $\delta_{ij}$ being the Kronecker delta and $A_{ij}$ encoding the edge weight from node $j$ to node $i$. An overall normalization factor is chosen so that the sum of all entries in $\mathbf{A}$, $\sum_{1 \le i,j \le n} A_{ij}$, equals $n-1$. As a consequence, $\frac{1}{n-1}\sum_{i=1}^{n} L_{ii}(t) = \frac{1}{n-1}\sum_{i=2}^{n} \lambda_i(t) = 1$, where the sum over the eigenvalues $\lambda_i(t)$ starts from $i=2$ because the trivial eigenvalue $\lambda_1$ associated with the eigenvector $\mathbf{v}_1 = (1, 1, \dots, 1)^\top$ is always 0. As a result of the normalization, the amount of resources (per node) used to maintain synchronization can be quantified solely by the coupling strength $\sigma$ for networks of different sizes and densities. The $d$-dimensional vector $\mathbf{x}_i$ describes the state of node $i$, $\mathbf{F}$ is the vector field dictating the intrinsic node dynamics, and $\mathbf{H}$ is the coupling function mediating interactions between different nodes.

To determine the stability of the synchronization state $\mathbf{x}_1(t) = \mathbf{x}_2(t) = \cdots = \mathbf{x}_n(t) = \mathbf{s}(t)$, we study the variational equation

$$\dot{\boldsymbol{\delta}} = \left[\mathbb{1}_n \otimes J\mathbf{F}(\mathbf{s}) - \sigma \mathbf{L}(t) \otimes J\mathbf{H}(\mathbf{s})\right]\boldsymbol{\delta}. \tag{2}$$

Here, $\boldsymbol{\delta} = (\mathbf{x}_1 - \mathbf{s}, \dots, \mathbf{x}_n - \mathbf{s})^\top$ is the perturbation vector, $\mathbb{1}_n$ is the $n \times n$ identity matrix, $\otimes$ represents the Kronecker product, and $J$ is the Jacobian operator. When the Laplacian matrices $\mathbf{L}(t)$ and $\mathbf{L}(t')$ commute for any $t$ and $t'$, following the master stability function formalism[18,29], we can find an orthogonal matrix $\mathbf{Q}$ such that $\mathbf{Q}^\top \mathbf{L}(t)\mathbf{Q}$ is diagonal for all time $t$, thus decoupling Eq. (2) into $n$ independent $d$-dimensional equations

$$\dot{\boldsymbol{\eta}}_i = \left[J\mathbf{F}(\mathbf{s}) - \sigma\lambda_i(t)J\mathbf{H}(\mathbf{s})\right]\boldsymbol{\eta}_i, \quad i = 1, \dots, n. \tag{3}$$

Here, $\{\boldsymbol{\eta}_i\}$ is linked to the original coordinates through the relation $(\boldsymbol{\eta}_1, \dots, \boldsymbol{\eta}_n)^\top = (\mathbf{Q}^\top \otimes \mathbb{1}_d)\boldsymbol{\delta}$. Each decoupled equation describes the evolution of an independent perturbation mode $\boldsymbol{\eta}_i$. In order for synchronization to be stable, all perturbation modes transverse to the synchronization manifold (namely, the modes $\boldsymbol{\eta}_2$ to $\boldsymbol{\eta}_n$) must asymptotically decay to zero. Since the decoupled variational equations are all of the same form and only differ in $\lambda_i(t)$, it is informative to study the maximum Lyapunov exponent of the equation

$$\dot{\boldsymbol{\xi}} = [J\mathbf{F}(\mathbf{s}) - \alpha J\mathbf{H}(\mathbf{s})]\boldsymbol{\xi} \tag{4}$$

as a function of $\alpha$. We refer to this function as the master stability function and denote it as $\Lambda(\alpha)$.

As we will show throughout the rest of the paper, if $\Lambda''(\alpha_0) < 0$ when $\Lambda(\alpha)$ first becomes negative at $\alpha_0 = \sigma_c$ (Fig. 1), then it is guaranteed that there exist temporal networks that outperform optimal static networks. Intuitively, this is because introducing temporal variation in the network structure allows all nonzero $\lambda_i(t)$ to spend a significant amount of time above 1, the optimal value achievable by static networks. (For static networks, because $\sum_{i=2}^{n} \lambda_i = n-1$, there must exist $0 < \lambda_i < 1$ unless all nonzero

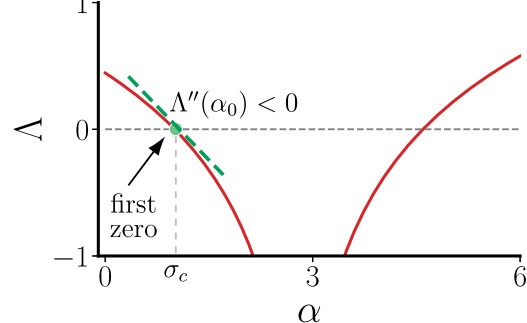

**Fig. 1 Curvature of the master stability function at its first zero.** Example master stability function for which temporal networks can synchronize stably below the critical coupling strength $\sigma_c$.

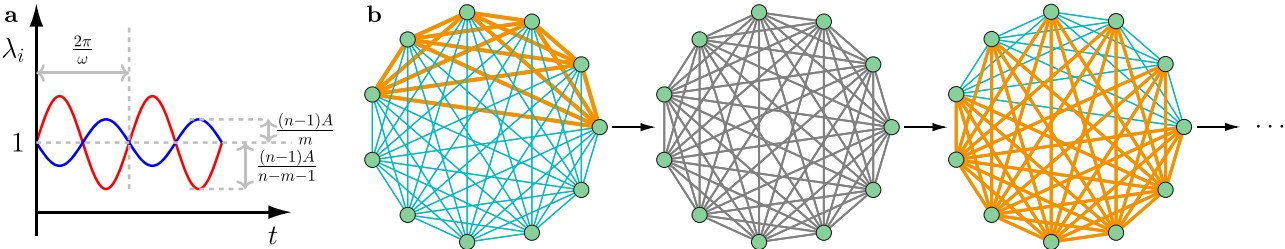

**Fig. 2 Designing temporal networks that synchronize better than optimal static networks. a** Evolution of the nonzero Laplacian eigenvalues described in Eq. (5), which are split into two degenerate groups. **b** Temporal network constructed from the Laplacian eigenvalues in **a**. The weight of each edge is represented by its thickness. In addition, edges whose weight is larger than $\frac{1}{n}$ are colored orange, whereas those with weight less than $\frac{1}{n}$ are colored cyan. For this network diagram, we set $n = 11$ and $m = 5$, and the corresponding weighted adjacency matrix is given by Eq. (9). Visually, we can see that different parts of the network are being strengthened in an alternating fashion.

eigenvalues are identical, in which case $\lambda_i = 1$ for all $i \geq 2$ and the network is optimal.) If $\Lambda''(\alpha_0) < 0$, the synchronization state can gain more stability while $\lambda_i(t) > 1$ than the stability it loses during the period when $\lambda_i(t) < 1$.

**Temporal networks that outperform optimal static networks.** In order to illustrate a simple scheme for designing temporal networks that synchronize for coupling strength below the critical value $\sigma_c$, we construct a class of Laplacian matrices that have the following spectrum (Fig. 2a):

$$\lambda_i(t) = \begin{cases} 0 & i = 1, \\ 1 + \frac{n-1}{m} A \sin(\omega t) & i = 2, \ldots, m+1, \\ 1 - \frac{n-1}{n-m-1} A \sin(\omega t) & i = m+2, \ldots, n. \end{cases} \quad (5)$$

The nonzero eigenvalues split into two groups with a time-varying gap between them, whereas their sum remains equal to $n-1$ for all time $t$. Intuitively, some of the perturbation modes borrow resources from the others to remain stable and then return the favor at a later time. As a result, this kind of dynamic stabilization achieves global synchronization with very limited resources.

One can design networks with a given spectrum by specifying a set of orthonormal eigenvectors $\{\mathbf{v}_i\}$[18]. For our purpose, any choice of $\{\mathbf{v}_i\}$ containing $\mathbf{v}_1 = (1, 1, \ldots, 1)^\top$ is valid, which gives rise to a whole range of synchronization-boosting temporal networks. Here, for concreteness, we adopt the eigenbasis proposed in ref. [31]:

$$\mathbf{v}_i = \left( \underbrace{\frac{1}{\sqrt{i(i-1)}}, \cdots, \frac{1}{\sqrt{i(i-1)}}}_{i-1 \text{ copies}}, -\frac{i-1}{\sqrt{i(i-1)}}, \underbrace{0, \cdots, 0}_{n-i \text{ copies}} \right)^\top,$$

$$(6)$$

where $i \geq 2$. Combining Eqs. (5) and (6) using the formula $\mathbf{L}(t) = \sum_{i=2}^{n} \lambda_i(t) \mathbf{v}_i \mathbf{v}_i^\top$ gives rise to a temporal network described by the following weighted adjacency matrix (Fig. 2b):

$$A_{ij}(t) = \begin{cases} \frac{\lambda_n(t)}{n} + \frac{\lambda_2(t) - \lambda_n(t)}{m+1} & i, j \leq m+1, i \neq j, \\ \frac{\lambda_n(t)}{n} & i \text{ or } j > m+1, i \neq j. \end{cases} \quad (7)$$

Substituting Eq. (5) into Eq. (7) shows that edges connecting the first $m + 1$ nodes have a time-dependent weight of $\frac{1}{n} + \frac{n(n-1)^2 - m(m+1)(n-1)}{nm(m+1)(n-m-1)} A \sin(\omega t)$, whereas the weight of the other edges evolves according to $\frac{1}{n} - \frac{(n-1)}{n(n-m-1)} A \sin(\omega t)$. The choice of the time-varying term $\sin(\omega t)$ is not essential; the sine function

can be replaced by any other periodic function $p(t)$ with period $T$ that satisfies $\int_0^T p(t)\, dt = 0$.

When assuming $n$ odd and $m = \frac{n-1}{2}$, we get a particularly simple class of temporal networks whose transverse perturbation modes all have the same stability (analogous to the defining property of optimal static networks):

$$\lambda_i(t) = \begin{cases} 0 & i = 1, \\ 1 + 2A \sin(\omega t) & i = 2, \ldots, \frac{n+1}{2}, \\ 1 - 2A \sin(\omega t) & i = \frac{n+3}{2}, \ldots, n, \end{cases} \quad (8)$$

$$A_{ij}(t) = \begin{cases} \frac{1 + (6 - \frac{8}{n+1})A \sin(\omega t)}{n} & i, j \leq \frac{n+1}{2}, i \neq j, \\ \frac{1 - 2A \sin(\omega t)}{n} & i \text{ or } j > \frac{n+1}{2}, i \neq j. \end{cases} \quad (9)$$

**Critical role of the switching rate.** To demonstrate the effectiveness of our design, we equip the temporal networks described by Eq. (9) with concrete node dynamics and probe their synchronizability in depth. Here, we choose Stuart–Landau oscillators as our first example, as they represent the canonical dynamics of systems in the vicinity of a Hopf bifurcation[32]. The oscillators evolve according to the following dynamical equation:

$$\dot{Z}_j = Z_j - (1 + ic_2)|Z_j|^2 Z_j - \sigma \sum_{k=1}^n L_{jk}(t)(1 + ic_1)Z_k, \quad (10)$$

where $Z_j = x_j + iy_j = r_j e^{i\theta_j} \in \mathbb{C}$ represents the state of the $j$th oscillator. Equation (10) is the discrete-space counterpart of the Ginzburg–Landau equation[33] and admits a limit-cycle synchronous state $Z_j(t) = e^{-ic_2 t} \forall j$. By writing the perturbations in polar coordinates, we find that the Jacobian terms in Eq. (4) become $J\mathbf{F} = \begin{pmatrix} -2 & 0 \\ -2c_2 & 0 \end{pmatrix}$ and $J\mathbf{H} = \begin{pmatrix} 1 & -c_1 \\ c_1 & 1 \end{pmatrix}$, both of which are constant matrices. Thus, according to Eq. (4), the master stability function can be obtained by solving a characteristic polynomial equation and has the following form[28]:

$$\Lambda(\alpha) = -\alpha - 1 + \sqrt{1 - 2c_1 c_2 \alpha - c_1^2 \alpha^2}. \quad (11)$$

Figure 3a shows $\Lambda(\alpha)$ for $c_1 = -1.8$ and $c_2 = 4$, which clearly has $\Lambda''(\alpha_0) < 0$ at its first zero $\alpha_0 \approx 3$.

For Stuart–Landau oscillators coupled on temporal networks, Eq. (3) dictates the stability of individual perturbation modes and can be written as

$$\dot{\boldsymbol{\eta}}_i = \mathbf{B}_i(t)\boldsymbol{\eta}_i, \quad (12)$$

where $\mathbf{B}_i(t) = \begin{pmatrix} -2 - \sigma\lambda_i(t) & c_1\sigma\lambda_i(t) \\ -2c_2 - c_1\sigma\lambda_i(t) & -\sigma\lambda_i(t) \end{pmatrix}$ is periodic with period $T = \frac{2\pi}{\omega}$ (henceforth, we drop the subscript $i$ to ease the notation). According to Floquet theory[34], the solution to Eq. (12)

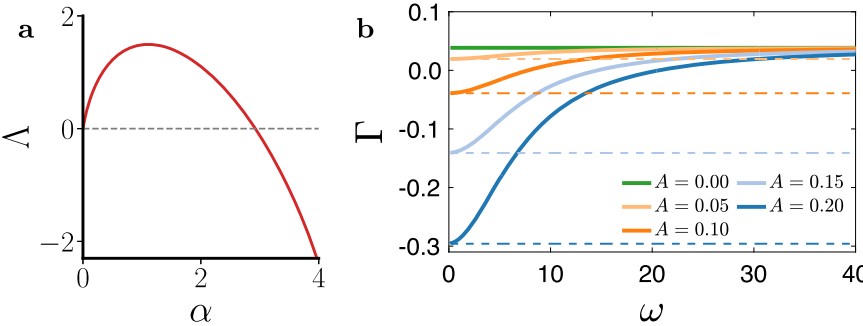

**Fig. 3 Temporal networks enable synchronization among Stuart–Landau oscillators. a** Master stability function for Stuart–Landau oscillators. Parameters are set to $c_1 = -1.8$ and $c_2 = 4$. **b** Maximum Lyapunov exponent $\Gamma$ as a function of the switching rate $\omega$ for different values of the temporal activity $A$ (solid lines). The dashed lines indicate the slow-switching limit predicted by the averaged master stability function $\bar{\Lambda}$.

must be of the form $e^{\mu t}\mathbf{P}(t)$, where $\mathbf{P}(t)$ has period $T$. The Floquet exponents $\mu_1$ and $\mu_2$ can be extracted by finding the principal fundamental matrix, and their real parts are the corresponding Lyapunov exponents[35]. Figure 3b shows the maximum Lyapunov exponent $\Gamma = \max\{\mathrm{Re}(\mu_1), \mathrm{Re}(\mu_2)\}$ as a function of $\omega$ for different values of the temporal activity $A$. (It is clear from Eq. (8) that all transverse perturbation modes have the same $\Gamma$. Thus, $\Gamma$ is also the maximum transverse Lyapunov exponent and determines the synchronization stability.) We set the coupling strength to slightly below $\sigma_c$ at $\sigma = 2.9$ so that no static network can synchronize. As the temporal activity $A$ is increased, $\Gamma$ becomes negative for an increasingly wide range of switching rate $\omega$, signaling that temporal variation in the network structure is successfully stabilizing synchronization under the given coupling budget.

Since the only difference between Eqs. (3) and (4) is the periodic $\lambda(t)$ vs. the fixed $\alpha$, it is natural to expect the stability of the temporal network to be related to the master stability function averaged over a suitable range of $\alpha$. Specifically, one might reasonably associate $\Gamma$ with the averaged master stability function[18,22,26,27,30]

$$\bar{\Lambda} = \int_{\lambda_{\min}}^{\lambda_{\max}} W(\lambda)\Lambda(\sigma\lambda)\,\mathrm{d}\lambda, \tag{13}$$

where $W(\lambda)$ is the probability distribution of $\lambda$ (it follows that $\int_{\lambda_{\min}}^{\lambda_{\max}} W(\lambda)\,\mathrm{d}\lambda = 1$). However, it is clear that $\bar{\Lambda}$ cannot be used to predict $\Gamma$ in general. One immediate observation is that $\bar{\Lambda}$ does not depend on the rate in which $\lambda(t)$ is changing (it only depends on the distribution of $\lambda$), whereas the curves representing $\Gamma$ in Fig. 3b clearly depend on the switching rate $\omega$. Indeed, in order to go from $\Gamma$ to $\bar{\Lambda}$, we are required to shuffle $\mathbf{B}(t)$ temporally in Eq. (12). This operation is forbidden when the matrices $\{\mathbf{B}(t)|t \in \mathbb{R}\}$ do not commute (or, equivalently, when $\{\mathbf{B}(t)|t \in \mathbb{R}\}$ cannot be simultaneously diagonalized). To see why, we can look at the formal solution to Eq. (12) expressed in terms of the matrix exponential:

$$\boldsymbol{\eta}(t) = \boldsymbol{\eta}(0)e^{\boldsymbol{\Omega}(t)}, \tag{14}$$

where $\boldsymbol{\Omega}(t)$ is given by the Magnus expansion[36]:

$$\boldsymbol{\Omega}(t) = \int_0^t \mathbf{B}(\tau)\,\mathrm{d}\tau + \frac{1}{2}\int_0^t \mathrm{d}\tau \int_0^\tau \mathrm{d}\tau'[\mathbf{B}(\tau), \mathbf{B}(\tau')]$$
$$+ \text{higher-order terms involving nested matrix commutators.} \tag{15}$$

Here, $[\mathbf{B}(\tau), \mathbf{B}(\tau')] = \mathbf{B}(\tau)\mathbf{B}(\tau') - \mathbf{B}(\tau')\mathbf{B}(\tau)$ is the matrix commutator. Equation (15) makes it clear that $\{\mathbf{B}(\tau)|0 < \tau < t\}$ can be shuffled without affecting $\boldsymbol{\Omega}(t)$ if and only if $[\mathbf{B}(\tau), \mathbf{B}(\tau')] = 0$ for all $\tau' < \tau < t$, in which case everything on the right-hand side except the first term vanishes.

However, $\bar{\Lambda}$ is still extremely informative on whether a given temporal network can synchronize or not. In particular, for $\omega \to 0$ (i.e., slow-switching networks[30]), $\Gamma$ approaches the value of $\bar{\Lambda}$, as demonstrated in Fig. 3b. Intuitively, this can be understood through a process we call "grow and rotate". When the matrices $\{\mathbf{B}(t)|t \in \mathbb{R}\}$ commute, $\boldsymbol{\eta}$ can be decomposed into components that grow independently along the eigendirections of $\mathbf{B}(t)$, whose growth rates are dictated by the corresponding eigenvalues. Eventually, the component along the direction with the largest eigenvalue becomes dominant. However, when $\{\mathbf{B}(t)|t \in \mathbb{R}\}$ do not commute, the growth along the eigendirections are often "interrupted", as the eigenvectors of $\mathbf{B}(t)$ are no longer fixed and will rotate over time. To keep track of the growth of the dominant component, we must project $\boldsymbol{\eta}$ onto the new dominant eigendirection upon rotation. These frequent projections can significantly influence the asymptotic growth rate (this is also why the maximum Lyapunov exponent is usually not the mean of the maximum local Lyapunov exponents). At the slow-switching limit, $\boldsymbol{\eta}$ can grow along an eigendirection uninterrupted for long enough that the effect of the projections becomes negligible. In this case, $\Gamma$ is determined by the average growth rate of $\boldsymbol{\eta}$ in the dominant direction of each $\mathbf{B}(t)$, which is exactly $\bar{\Lambda}$.

It is worth noting that the equivalence between $\Gamma$ and $\bar{\Lambda}$ at the slow-switching limit is not specific to Stuart–Landau oscillators and can be expected for generic oscillator models[26,30]. As a result, $\Lambda''(\alpha_0) < 0$ is a robust indicator that synchronization in a system can benefit from temporal networks. This observation echoes recent results in ref. [30], which demonstrates the importance of a master stability function's curvature for synchronization in the special case of networks with fixed topology and time-varying overall coupling strength. To see why curvature has such a critical role, we assume the temporal variation of $\lambda$ around 1 to be small and Taylor expand $\Lambda(\alpha)$ around $\alpha_0$. Then, the averaged master stability function for coupling strength $\sigma = \sigma_c$ is

$$
\begin{aligned}
\bar{\Lambda} &= \int_{1-\epsilon}^{1+\epsilon} W(\lambda)\Lambda(\sigma_c\lambda)\,\mathrm{d}\lambda \\
&= \int_{1-\epsilon}^{1+\epsilon} W(\lambda)\left[\Lambda(\sigma_c) + \Lambda'(\sigma_c)(\lambda - 1) + \frac{1}{2}\Lambda''(\sigma_c)(\lambda-1)^2\right]\mathrm{d}\lambda + \mathcal{O}(\epsilon^3) \\
&= \underbrace{\Lambda(\sigma_c)}_{=0} + \Lambda'(\sigma_c)\underbrace{\int_{1-\epsilon}^{1+\epsilon} W(\lambda)(\lambda - 1)\mathrm{d}\lambda}_{=0} + \frac{1}{2}\Lambda''(\sigma_c)\underbrace{\int_{1-\epsilon}^{1+\epsilon} W(\lambda)(\lambda - 1)^2\mathrm{d}\lambda}_{>0} + \mathcal{O}(\epsilon^3).
\end{aligned}
\tag{16}
$$

Thus, if $\Lambda''(\alpha_0) = \Lambda''(\sigma_c) < 0$, then $\bar{\Lambda} < 0$ at $\sigma = \sigma_c$ and stability is guaranteed to be improved at the slow-switching limit, where $\Gamma = \bar{\Lambda}$. This improvement is expected to extend into the intermediate switching rate due to the continuity of $\Gamma$ as a function of $\omega$.

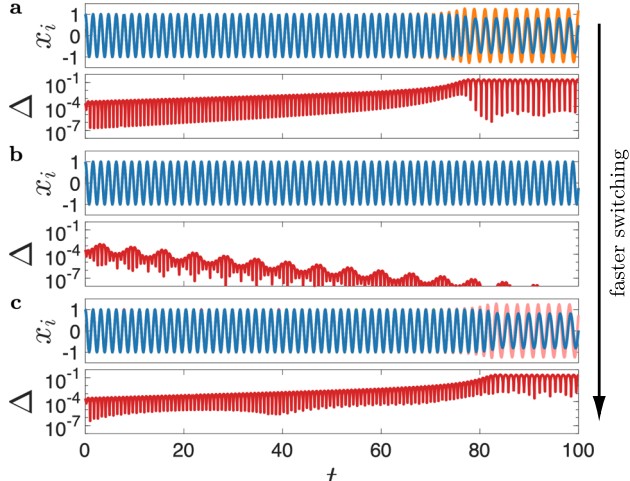

**Fig. 4 Temporal networks that synchronize only for intermediate switching rate.** Evolution of the oscillator states $x_i$ and the synchronization error $\Delta$ for **a**: $\omega = 0$, **b**: $\omega = 1$, and **c**: $\omega = 100$. The oscillator parameters are the same as in Fig. 3 and the underlying temporal network is illustrated in Fig. 2b.

At the other limit, for $\omega \to \infty$ (i.e., fast-switching networks), $\Gamma$ clearly does not match with $\bar{\Lambda}$. In particular, $\Gamma$ does not depend on the temporal activity $A$. For the system in Fig. 3b, $\Gamma$ approaches $\Lambda(\sigma)$ as $\omega \to \infty$, which is the value expected for an optimal static network at coupling strength $\sigma$ (in this case the time-averaged network is a complete graph with uniform edge weights). The mapping from a temporal network to its time-averaged counterpart at the fast-switching limit is intuitive and well established in the literature[17,23,25].

The results above provide new insights into the intriguing phenomenon that certain temporal networks only synchronize for intermediate switching rate[21,26,27]: when switching is too fast, the temporal network reduces to its static counterpart and one cannot take full advantage of the temporal variation in the connections; when switching is too slow, although the asymptotic stability might be maximized, the system would have desynchronized long before the network experiences any meaningful change. Thus, the sweet spot often emerges at an intermediate switching rate.

In Fig. 4, we show typical trajectories of $n = 11$ Stuart–Landau oscillators on the temporal networks described by Eq. (9), with the temporal activity set to $A = 0.15$. Systems in all three panels are initiated close to the synchronous state, and their only difference lies in the switching rate $\omega$, which allows us to compare networks with static, moderate-switching, and fast-switching topologies. By monitoring the synchronization error $\Delta(t)$, defined as the standard deviation among $Z_j(t)$, we see that only the system with an intermediate switching rate ($\omega = 1$, panel b) can maintain stable synchrony. Interestingly, $\Delta(t)$ in that system goes down non-monotonically and is bounded from above by periodic envelopes. The width of each envelope is $2\pi$, which coincides with the period of the changing network topology.

**Universal stabilization of low-dimensional maps**. The framework developed so far can be readily transferred from differential equations to discrete maps, from continuous variation in the network topology to discrete switching, and from periodic oscillator dynamics to chaotic ones. The discrete-time analog of Eq. (1) can be written as

$$\mathbf{x}_i[t+1] = \beta\mathbf{F}(\mathbf{x}_i[t]) - \sigma\sum_{j=1}^{n}L_{ij}[t]\mathbf{H}(\mathbf{x}_j[t]), \quad i = 1, \dots, n. \quad (17)$$

To demonstrate the advantage of temporal networks in these settings, we focus on the following class of coupled one-dimensional discrete maps:

$$x_i[t+1] = \beta F(x_i[t]) - \sigma\sum_{j=1}^{n}L_{ij}[t]F(x_j[t]), \quad i = 1, \dots, n, \quad (18)$$

where $F : \mathbb{R} \to \mathbb{R}$ is the mapping function. As we show below, this setup allows us to develop an elegant theory that offers new insights.

Similar to the continuous-time case, the synchronization stability is determined by the decoupled variational equations

$$\eta_i[t+1] = \left[(\beta - \sigma\lambda_i[t])F'(s[t])\right]\eta_i[t], \quad i = 2, \dots, n. \quad (19)$$

For fixed $\lambda$, the Lyapunov exponent of Eq. (19) is given by $\ln|\beta - \sigma\lambda| + \Gamma_s$, where $\Gamma_s = \lim_{\mathcal{T}\to\infty}\frac{1}{\mathcal{T}}\sum_{t=1}^{\mathcal{T}}\ln|F'(s[t])|$ is a finite constant. Thus, the master stability function has a universal form (illustrated in Fig. 1)

$$\Lambda(\alpha) = \ln|\alpha - \beta| + \Gamma_s. \quad (20)$$

Taking the second derivative with respect to $\alpha$, we see that

$$\Lambda''(\alpha) = -\frac{1}{(\alpha - \beta)^2} < 0. \quad (21)$$

Thus, synchronization in any system described by Eq. (18) can benefit from the temporal networks designed in this paper. In particular, this holds for any mapping function $F$, which encompasses important dynamical systems such as logistic maps, circle maps, and Bernoulli maps.

For concreteness, we set $F(x) = \sin^2(x + \pi/4)$ and $\beta = 2.8$ (the corresponding $\Gamma_s = -0.5855$), which models the dynamics of coupled optoelectronic oscillators[37] and exhibits chaotic dynamics. The time-discretized version of the temporal networks described by Eq. (7) works out-of-the-box for the optoelectronic oscillators, despite the vastly different node dynamics. Here, to demonstrate the flexibility of our network design, we consider the following slightly modified switching scheme, which is also more natural for discrete-time systems:

$$A_{ij}[t] = \begin{cases} \frac{1+(-1)^{\lfloor t/T\rfloor}(6-\frac{8}{n+1})A}{n} & i,j \le \frac{n+1}{2}, i\neq j, \\ \frac{1-(-1)^{\lfloor t/T\rfloor}2A}{n} & i \text{ or } j > \frac{n+1}{2}, i\neq j, \end{cases} \quad (22)$$

where $\lfloor\cdot\rfloor$ is the floor function. Basically, the network switches between two configurations every $T$ iterations, with each configuration being the extremal in the continuous scheme described by Eq. (9). Consequently, every nonzero eigenvalue of the temporal Laplacian alternates between $1 + 2A$ and $1 - 2A$ with period $T$.

Again, the averaged master stability function $\bar{\Lambda}$ accurately predicts the stability of the temporal network at the slow-switching limit. More interestingly, for systems described by Eq. (18), the connection is much stronger: $\bar{\Lambda}$ determines the stability of the temporal network for all switching periods $T$. To see why, we note that the synchronization stability is determined by the limit product $\prod_{t=1}^{\infty}(\beta - \sigma\lambda[t])F'(s[t])$. Normally, these are matrix products and cannot be reordered. However, since $1 \times 1$ matrix multiplications commute, for one-dimensional maps we can

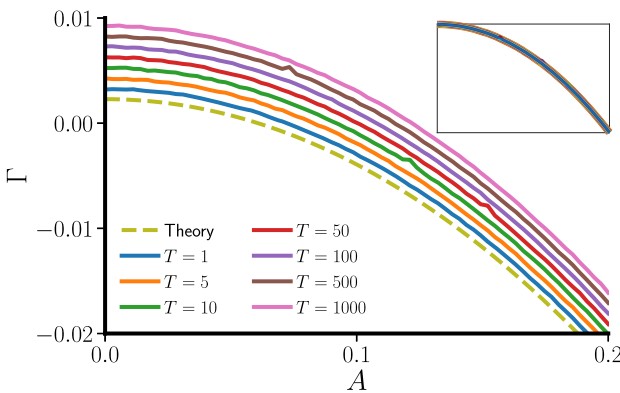

**Fig. 5 Temporal networks promote synchronization universally in discrete-time systems with low-dimensional node dynamics.** The maximum transverse Lyapunov exponent $\Gamma$ decreases as the temporal activity $A$ is increased. The dashed line represents the theoretical prediction based on $\bar{\Lambda}$, whereas the solid lines (shifted vertically for visibility) are calculated directly from Eq. (19) for different switching periods $T$. Without the shift, all curves completely overlap (inset), which confirms our prediction that the stabilization provided by temporal networks does not depend on the switching rate for coupled one-dimensional maps.

reorder them to obtain

$$
\begin{aligned}
\Gamma &= \lim_{\mathcal{T}\to\infty} \frac{1}{\mathcal{T}} \ln \left| \prod_{t=1}^{\mathcal{T}} (\beta - \sigma\lambda[t]) F'(s[t]) \right| \\
&= \int_{\lambda_{\min}}^{\lambda_{\max}} W(\lambda) \ln |\beta - \sigma\lambda| \, \mathrm{d}\lambda + \lim_{\mathcal{T}\to\infty} \frac{1}{\mathcal{T}} \sum_{t=1}^{\mathcal{T}} \ln |F'(s[t])| \\
&= \int_{\lambda_{\min}}^{\lambda_{\max}} W(\lambda) \left( \ln |\beta - \sigma\lambda| + \Gamma_s \right) \mathrm{d}\lambda \\
&= \int_{\lambda_{\min}}^{\lambda_{\max}} W(\lambda) \Lambda(\sigma\lambda) \, \mathrm{d}\lambda = \bar{\Lambda}.
\end{aligned}
\tag{23}
$$

This independence of $\Gamma$ on $T$ might seem contradictory to the fact that, at the fast-switching limit, temporal networks can be reduced to their static counterparts. But notice that there is usually no fast switching in discrete-time systems—even if the network topology changes at every iteration, it is still evolving at the same timescale as the node dynamics. Moreover, unlike in continuous-time systems[16,17,23,25], the discrete nature of the dynamics precludes the use of the averaging techniques[16,25] essential for connecting fast-switching networks with their time-averaged counterparts. Thus, one cannot map a temporal network to its time-averaged counterpart in discrete-time systems even when the network topology changes much more rapidly than the node dynamics.

In Fig. 5, we show the maximum transverse Lyapunov exponent $\Gamma$ of the synchronization state in the optoelectronic system for $\sigma = 1$, which is slightly below $\sigma_c$. The dashed line corresponds to the theoretical prediction of $\Gamma$ based on the averaged master stability function $\bar{\Lambda} = \frac{1}{2} \left( \ln |1 + 2A - \beta| + \ln |1 - 2A - \beta| \right) + \Gamma_s$. As expected, the static network ($A = 0$), despite being optimal, is unstable. As the temporal activity $A$ is increased, $\overline{\Lambda}$ deceases and synchronization is eventually stabilized. On the other hand, the solid lines represent $\Gamma$ obtained numerically by evolving Eq. (19) for different switching periods $T$. These lines are shifted vertically by different amounts in Fig. 5, purely as an aid to the eye. The unshifted versions are shown in the inset. Notice that all the lines collapse onto a single curve, demonstrating the excellent agreement between theory and simulations.

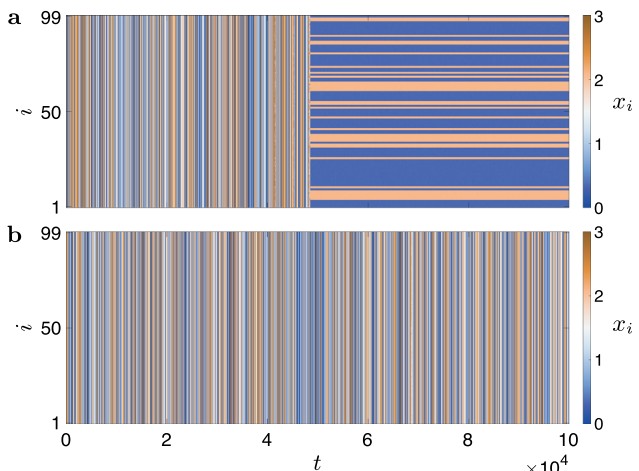

**Fig. 6 Improved synchronization in aperiodic and noncommutative temporal networks.** The temporal networks are based on the discrete-switching networks given by Eq. (22), which are further made aperiodic and noncommutative by applying random Gaussian perturbations of zero mean to the strength of each edge independently at every time step $t$. The standard deviation of the perturbations is fixed at $0.1/n$ (10% of the average edge weight). The spacetime plots show the evolution of the optoelectronic oscillators on temporal networks with **a**: temporal activity $A = 0$ and **b**: temporal activity $A = 0.15$. Both systems are initialized close to the synchronous state. Synchronization persists only in the second system, even though the network in the first system is an optimal static network (a complete graph with uniform edge weights). Other parameters are set to $n = 99$, $T = 10$, $\beta = 2.8$, and $\sigma = 1.05$.

An interesting question is what happens when we introduce random fluctuations to the network structure at each time step $t$, which makes the temporal network aperiodic and the graph Laplacians noncommutative. In Fig. 6, through direct simulations[35], we show that temporal networks still outperform optimal static networks in the presence of these random fluctuations. Here, we use the same model of optoelectronic oscillators and the discrete-switching network considered in Fig. 5, except that independent random Gaussian perturbations of zero mean and standard deviation $0.1/n$ (10% of the average edge weight) are added to the strength of each edge at every time step. For temporal activity $A = 0$ (Fig. 6a), synchronization cannot be sustained at coupling strength $\sigma = 1.05$. For temporal activity $A = 0.15$ (Fig. 6b), synchronization is stabilized at the same coupling strength by the variation in network structure. The network size is set to $n = 99$ and the switching period to $T = 10$ in our simulations, although the results do not depend sensitively on these two parameters.

## Discussion

To summarize, we have designed temporal networks that synchronize more efficiently than optimal static networks. These temporal networks are particularly relevant when the coupling budget available in a system to maintain stable synchrony is limited. We provided analytical insight into the synchronizability of commutative temporal networks by linking it to the curvature of the corresponding master stability function. In particular, our analysis reveals the subtle relation between the performance of a temporal network and its switching rate. The switching rate has an especially critical role in systems with high-dimensional oscillator dynamics, and networks with intermediate switching rates often emerge as the most effective.

Our open-loop design has several advantages compared with closed-loop schemes where the network structure is adjusted

on-the-fly based on feedback from the node states (often modeled by adaptive networks[38]). First, our design does not depend sensitively on the node dynamics. As we have shown, the same design works for systems with vastly different node dynamics, and it applies readily to both continuous-time and discrete-time systems. Second, we do not need to monitor all the nodes constantly, which also eliminates the possibility of being detrimentally influenced by measurement errors. Third, the evolution of the network is highly predictable and we can easily control the coupling budget allocated to the system at any given time $t$, a task that is far more difficult in adaptive networks. On the other hand, closed-loop schemes have the advantage of being readily adaptive to the changing environment and can react quickly to unexpected perturbations[39,40]. A promising future direction would be to devise hybrid schemes that combine the best from both worlds, which could enable even more efficient and robust synchronization.

In this work, for the sake of analytical tractability, we mostly focused on temporal networks whose Laplacian matrices from different time instants commute. There is evidence that synchronization in temporal networks can benefit when $\mathbf{L}(t)\mathbf{L}(t')\neq\mathbf{L}(t')\mathbf{L}(t)$[20]. It would therefore be interesting to see whether our design of temporal networks could be further optimized by allowing noncommuting Laplacian matrices. In particular, can random fluctuations in the network structure (which give rise to noncommuting Laplacian matrices in general) outperform our designed temporal networks? More generally, do optimal temporal networks exist for the purpose of synchronization, just like there are optimal static networks? And if so, what are their defining characteristics?

Finally, we hope our results can serve as an important step towards achieving efficient synchronization in complex interconnected systems. For example, many temporal networks arise naturally in the real world through moving agents, whose interactions depend on their spatial distance[41–44]. An exciting next step is to understand how our design can be implemented in such systems and how the time-varying connections can be translated into the spatial movement of individual agents.

## Data availability
All data needed to evaluate the conclusions are presented in the paper. Additional data related to this paper may be requested from the authors.

## Code availability
Code for performing network dynamics simulations and stability calculations are available at https://github.com/y-z-zhang/temporal_sync. An archived version of the code is also provided[35]. Additional source code may be requested from the authors.

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

## Acknowledgements

Y.Z. acknowledges support from the Schmidt Science Fellowship.

## Author contributions

Y.Z. and S.H.S. conceived the project. Y.Z. performed the research. Y.Z. wrote the paper with input from S.H.S.

## Competing interests

The authors declare no competing interests.
