## [Peer Review File · Nature Communications]

Reviewers' Comments:

Reviewer #1:

Remarks to the Author:

The manuscript "Designing temporal networks that synchronize under resource constraints" represents a clear step forward in our understanding about dynamical properties of complex systems on top of time dependent networks.

Although initially the authors formulate as a constraint the "amount of coupling" the development they perform approaches the problem of time dependent networks. The dependence in the topology, or in general in the coupling, is tackled by means of generalization of the master stability function approach providing analytically the stability conditions. Under an oscillating "weighted adjacency matrix" they show that a better synchronizability is obtained, this ensuring that the "cost" in terms of total amount of coupling intensity can be minimized and synchronization achieved. The authors show that this approach is also valid for low-dimensional maps.

As the authors state, this approach could also be used in the case of moving agents which represents a particular time dependent topology and coupling. In this case coupling exchange is limited to agents that are within a defined range, there is no oscillating rule, and in principle it is not clear whether the same rule could be applied. In any case, this is a wide open field of research to which the the paper under consideration makes a very promising contribution.

The paper is very well written and presented, and there is no need of modifications, and for that reason I recommend the paper to be published as it is.

Reviewer #2:

Remarks to the Author:

The paper "Designing temporal networks that synchronize under resource constraints" by Zhang and Strogatz investigates the conditions that yield synchronization for a collection of non linear oscillators diffusively coupled via time varying networks. As opposed to previous analysis that focused on the fast-switching limit, the authors here link the stabilization to the curvature of the master stability function. This enables one to gain analytical insight into the synchronization on temporal networks in general and for intermediate switching rates. The paper is well written and the topic worth investigation. In carrying out the analysis the authors assume temporal networks whose Laplacian matrices from different time instants commute. This is a rather strong assumption and I think that the authors could consider, as an optional revision, to elaborate further on viable generalization beyond this specific setting. A possibility could be to follow the procedure discussed in the SI of reference [24]. The obtained results are however sound and clearly presented and, in this referee's opinion, the paper can be accepted for publication in Nat. Comm.

Reviewer #3:

Remarks to the Author:

Science is a collective process that involves the work of many. Historically, the great scientists who have had the fortune of "sitting on the shoulders of giants" have acknowledged essential contributions by their predecessors. The collective nature of the process by which Science evolves imposes on us that we follow certain rules, which include giving credit in our publications to those who have done work closely related to ours. The introduction is the proper section for placing a paper in the context of the existing literature and for the authors to construct the case for why their work deserves publication, which often involves highlighting differences with relevant previous work.

There are several reasons why explicit, clear, and fair references to relevant previous work are essential to any scientific publication and among these: (i) it helps placing the paper in the context of the existing literature, which is in the interest of the reader, (ii) it is a form of respect for the other scientists who have preceded us, by acknowledging that the new research is in one way or another "aware" of their previous work and (iii) it provides a clear understanding of the merit and

novelty of the new work that is presented to reviewers and editors.

When reading the paper here considered for publication, it struck me how closely related it is to this other paper published in 2006:

"Synchronization in dynamical networks: Evolution along commutative graphs", PHYSICAL REVIEW E 74, 016102 2006.

The commutative graphs are exactly the networks with commuting Laplacian matrices that the new paper focuses on. To provide an idea of how closely related the 2006 paper is, I copy below the abstract:

"Starting from an initial wiring of connections, we show that the synchronizability of a network can be significantly improved by evolving the graph along a time dependent connectivity matrix. We consider the case of connectivity matrices that commute at all times, and compare several approaches to engineer the corresponding commutative graphs. In particular, we show that synchronization in a dynamical network can be achieved even in the case in which each individual commutative graphs does not give rise to synchronized behavior."

It is thus clear that this 2006 paper represents the "key reference" for the new paper. This would have required a thorough discussion of how the new work relates to this closely related previous work, which instead is omitted. The only citation is presented on line 44, where the relevant reference [18] is briefly mentioned together with a series of other papers, but no specific discussion relative to [18] is presented. The other papers in the list are also relevant and worth mentioning but they do not constitute "key references", the same way that [18] does. In fact, the master stability function reduction considered, Eq. (3), based on the commuting Laplacians, is the same as that of [18] but not the same as that in the remaining references. However, following line 44, the discussion focuses on the other less relevant references, and so on the fast-switching limit discussed in [17, 23, 24]. No acknowledgement is presented of the very close connection with ref. [18] and no explanation for how the new work compares to [18]. Yet the authors must be aware of the paper, as the citation demonstrates. I would also argue that the fact that after 15 years new research so closely related to [18] is still being carried out and considered for publication in an important journal such as Nature Communications (2021 impact factor 12.3 compared to PRE impact factor 2.3) makes ref. [18] especially commendable and so even more worth of acknowledgment.

The problem becomes more serious when considering that the audience of Nature Communications is much broader than that of PRE. From the journal webpage one reads: "Nature Communications is an open access, multidisciplinary journal dedicated to publishing high-quality research in all areas of the biological, health, physical, chemical and Earth sciences." Therefore, the typical readership of this journal is not formed of nonlinear dynamicists. Were the paper published in Nature Communications in the current form, these non-expert readers would be induced to believe that the new paper is the first one to study the case of temporal networks evolving via commuting Laplacians (nowhere in the paper a reference to previous work on this point is included), which is clearly not the case. The lack of clear and fair references and comparisons is concerning. Once one is made aware of the previous work, the interest and novelty of the new work is strongly reduced, as it then becomes apparent that the contribution of the new work is incremental. Further technical comments are provided below:

1. Line 69: The Laplacian matrix is normalized by the average indegree... What does this mean mathematically? An equation should have been included to clarify this critical aspect. Note that this is also important as it indicates that the authors are looking at a very specific class of networks (different from Ref. 18)
2. The relation between Eq. (3) and (4) is not fully elucidated. What happens if the eigenvalues $\lambda_i(t)$ converge to zero or diverge? What happens then to the Lyapunov exponents of (3)? Does the theory in the paper only work for periodically time varying $\lambda_i(t)$, which is the case of all the examples presented? It is also stated on line 124 that the function $\sin(\omega t)$ can be replaced by any periodic function with zero time-average. Is this another limitation of the work? If

$\lambda_i(t)$ is forced to oscillate in some specific way, why isn't the true maximum Lyapunov exponent of Eq.(3) considered?

3. What if $\lambda_i(t)$ oscillates to get past the second zero of the master stability function in figure 1 or just to exit the immediate neighborhood of the first zero? Are the conclusion only supported for the case for which $\lambda_i(t)$ has small oscillations about the first zero?

4. The case in which the switching is too slow (in continuous time) or too fast (in discrete time) are dismissed too quickly. There are papers on slow switching that also would need to be referenced. In the continuous time case, it is stated that the network would desynchronize well before any meaningful change occurs, which I think is a reference to a nonlinear effect (but a fair comparison would need to be carried in the linear regime.) For discrete time systems it is stated that there is no fast switching – to which I would object that it is contingent on the specific form of the discrete time system.

5. Finally I am confused by the main conclusion that temporal networks can synchronize even when stable synchrony is "impossible in any static network under the given budget". The actual result is that the temporal network can synchronize if it spends some time in the region in which the Lyapunov exponent is negative (to the right of the first zero in fig. 1) and some time in the region in which the Lyapunov exponent is positive (to the left of the first zero in fig. 1.) Thus the condition for the temporal network to synchronize is that at a certain time it must be "equal" to a static network that can synchronize. Or does that individual instance not satisfy the budget? I find the whole story of the budget, without a proper mathematical definition, very confusing. The word budget is also used in the introduction to indicate the control energy, which is a difference concept.

To conclude, my recommendation is to resubmit this work to a more technical journal by making sure to include clear references to relevant previous work on temporal networks with commuting Laplacians and to emphasize the differences of the current work from this previous fundamental work.

Just to be clear: the matter is not just that a comparison with previous work is missing, it is that a comparison with a key reference is missing. After properly placing the paper in the context of the existing literature, it becomes apparent that the main contribution of the paper is incremental and much more limited than what one would conclude otherwise.

We thank the referees for their insightful and constructive comments. For the convenience of the referees, we include in this submission a marked PDF of the paper in which the main changes are highlighted in blue.

Responses to Specific Comments of Reviewer #1

Reviewer #1: *The manuscript "Designing temporal networks that synchronize under resource constraints" represents a clear step forward in our understanding about dynamical properties of complex systems on top of time dependent networks. Although initially the authors formulate as a constraint the "amount of coupling" the development they perform approaches the problem of time dependent networks. The dependence in the topology, or in general in the coupling, is tackled by means of generalization of the master stability function approach providing analytically the stability conditions. Under an oscillating "weighted adjacency matrix" they show that a better synchronizability is obtained, this ensuring that the "cost" in terms of total amount of coupling intensity can be minimized and synchronization achieved. The authors show that this approach is also valid for low-dimensional maps.*

As the authors state, this approach could also be used in the case of moving agents which represents a particular time dependent topology and coupling. In this case coupling exchange is limited to agents that are within a defined range, there is no oscillating rule, and in principle it is not clear whether the same rule could be applied. In any case, this is a wide open field of research to which the the paper under consideration makes a very promising contribution.

The paper is very well written and presented, and there is no need of modifications, and for that reason I recommend the paper to be published as it is.

Response: We thank the referee for the positive assessment of our paper and his/her appreciation of our work.

We agree that many types of moving agents only interact within a predefined range (and the cutoff is sharp). However, there are also agents whose interactions vary continuously with distance. Even in the former case, the discrete-switching networks designed by us [e.g., Eq. (22)] may still be applicable. Thus, we are hopeful that one day our results can inspire new strategies for synchronizing moving agents.

Responses to Specific Comments of Reviewer #2

Reviewer #2: *The paper “Designing temporal networks that synchronize under resource constraints” by Zhang and Strogatz investigates the conditions that yield synchronization for a collection of non linear oscillators diffusively coupled via time varying networks. As opposed to previous analysis that focused on the fast-switching limit, the authors here link the stabilization to the curvature of the master stability function. This enables one to gain analytical insight into the synchronization on temporal networks in general and for intermediate switching rates. The paper is well written and the topic worth investigation. In carrying out the analysis the authors assume temporal networks whose Laplacian matrices from different time instants commute. This is a rather strong assumption and I think that the authors could consider, as an optional revision, to elaborate further on viable generalization beyond this specific setting. A possibility could be to follow the procedure discussed in the SI of reference [24]. The obtained results are however sound and clearly presented and, in this referee’s opinion, the paper can be accepted for publication in Nat. Comm.*

Response: We thank the referee for the positive assessment of our paper and the constructive comments.

To demonstrate that our results extend beyond commutative graphs, we now add random fluctuations to the discrete-switching network studied in Fig. 5. The random fluctuations in network structure break the commutativity condition for the graph Laplacians. As we show in the new Fig. 6, our designed temporal networks continue to outperform optimal static networks in the presence of these random fluctuations.

Responses to Specific Comments of Reviewer #3

Reviewer #3:

...

When reading the paper here considered for publication, it struck me how closely related it is to this other paper published in 2006:

“Synchronization in dynamical networks: Evolution along commutative graphs”, PHYSICAL REVIEW E 74, 016102 2006.

...

It is thus clear that this 2006 paper represents the “key reference” for the new paper. This would have required a thorough discussion of how the new work relates to this closely related previous work, which instead is omitted. The only citation is presented on line 44, where the relevant reference [18] is briefly mentioned together with a series of other papers, but no specific discussion relative to [18] is presented. The other papers in the list are also relevant and worth mentioning but they do not constitute “key references”, the same way that [18] does. In fact, the master stability function reduction considered, Eq. (3), based on the commuting Laplacians, is the same as that of [18] but not the same as that in the remaining references. However, following line 44, the discussion focuses on the other less relevant references, and so on the fast-switching limit discussed in [17, 23, 24]. No acknowledgement is presented of the very close connection with ref. [18] and no explanation for how the new work compares to [18].

...

Were the paper published in Nature Communications in the current form, these non-expert readers would be induced to believe that the new paper is the first one to study the case of temporal networks evolving via commuting Laplacians (nowhere in the paper a reference to previous work on this point is included), which is clearly not the case. The lack of clear and fair references and comparisons is concerning. Once one is made aware of the previous work, the interest and novelty of the new work is strongly reduced, as it then becomes apparent that the contribution of the new work is incremental.

Response: We thank the referee for the careful review of the original manuscript and the significant energy he/she spent in writing this report. We have made an effort to address all aspects of the referee’s remarks in our revisions and responses below.

In particular, we have now highlighted the 2006 PRE paper (Ref. [18]). We mention its use of commutative graphs prominently in the Introduction (e.g, in Lines 60-63) and provide a clear comparison between Ref. [18] and our paper.

Below, we would like to further clarify the contribution of our work. To make things crystal clear, please allow us first to explain what is NOT novel in our manuscript. The following two points are NOT novel:

- The demonstration that temporal networks can synchronize better than *certain* static networks (e.g., their time-averaged or time-frozen counterparts).
- The use of commutative graphs in our formulation.

The first point was already demonstrated in numerous previous publications, including Ref. [18], as we noted in the Introduction of the original manuscript.

The second point is an assumption that we adopted for the sake of analytical tractability. As the referee correctly pointed out, this assumption was widely adopted in the literature, including in Ref. [18]. We agree with the referee that this aspect of Ref. [18] should be more explicitly acknowledged, since although it is obvious to us that multiple commutative matrices can be simultaneously diagonalized, it may not be to some of the readers of Nature Communications. Aside from clear attribution in the Introduction, we have now cited Ref. [18] at multiple places where the commutativity assumption is invoked (e.g., Lines 90 and 122). We want to assure the referee that it was never our intention to claim novelty for studying commutative graphs, whose use is a common practice in the nonlinear dynamics community (see, for example, Refs. [22, 25, 27]).

So, what are the key contributions of our manuscript?

- We established an analytical condition (in terms of the curvature of the master stability function) on when temporal networks are expected to promote synchronization and showed that it is satisfied by a wide range of systems.
- We designed an open-loop control scheme for generating temporal networks that synchronize better than ALL static networks when the total coupling budget is limited, demonstrating a fundamental advantage of temporal networks.

These two points represent significant advances of our understanding of synchronization on temporal networks, especially given that previous papers on this topic (including Ref. [18]) rarely consider the effect of resource constraints, an important aspect of many real-world systems.

Thus, we disagree with the referee's assessment that "Once one is made aware of the previous work, the interest and novelty of the new work is strongly reduced, as it then becomes apparent that the contribution of the new work is incremental." The insights presented in the 2006 PRE paper, although important, in no way compromise the novelty of our work.

Reviewer #3: *Further technical comments are provided below:*

1. Line 69: The Laplacian matrix is normalized by the average indegree... What does this mean mathematically? An equation should have been included to clarify this critical aspect. Note that this is also important as it indicates that the authors are looking at a very specific class of networks (different from Ref. 18)

Response: We agree that the term “average indegree” is not very precise, for this reason we have revised that part of the text to further clarify the normalization process. More details are also added to the mathematical definition on Lines 74 to 77.

Note that this normalization is by no means a limitation of our work and is only introduced to facilitate fair comparison between different networks in terms of the coupling resources they use. As we further explained on Lines 79 to 82, “As a result of the normalization, the amount of resources (per node) used to maintain synchronization can be quantified solely by the coupling strength σ for networks of different sizes and densities.”

Reviewer #3: *2. The relation between Eq. (3) and (4) is not fully elucidated. What happens if the eigenvalues $\lambda_i(t)$ converge to zero or diverge? What happens then to the Lyapunov exponents of (3)? Does the theory in the paper only work for periodically time varying $\lambda_i(t)$, which is the case of all the examples presented? It is also stated on line 124 that the function $\sin(\omega t)$ can be replaced by any periodic function with zero time-average. Is this another limitation of the work? If $\lambda_i(t)$ is forced to oscillate in some specific way, why isn't the true maximum Lyapunov exponent of Eq.(3) considered?*

Response: Since the total coupling strength is bounded, no eigenvalue will diverge to infinity. An eigenvalue can in principle reach zero at certain time instants, which does not pose any problem as far as we can tell. The Lyapunov exponents of Eq. (3) depends on the full evolution history of the eigenvalue.

Our theory works for aperiodic temporal networks as well. In the new Fig. 6, we show that our temporal networks outperform optimal static networks even when the network structure is subject to random fluctuations at each time step, which makes the network aperiodic. Moreover, the graph Laplacians in these perturbed temporal networks no longer commute, so Fig. 6 also demonstrates that our results are not limited to commutative graphs.

The zero time-average on Line 133 was introduced due to the constraint on the coupling budget. Intuitively, it means each perturbation mode needs to return the same amount of resource it borrowed in the past. This condition is a strength, not a weakness of our formulation. (Similar to an optimization problem, adding constraints makes the problem harder, not easier.)

We are a bit confused by the referee's last question, since the true maximum Lyapunov exponent of Eq. (3), Γ , is already shown in Figs. 3b and 5.

Reviewer #3: *3. What if $\lambda_i(t)$ oscillates to get past the second zero of the master stability function in figure 1 or just to exit the immediate neighborhood of the first zero? Are the conclusion only supported for the case for which $\lambda_i(t)$ has small oscillations about the first zero?*

Response: Our results are not limited to oscillations in the vicinity of the first zero. As clearly demonstrated in Figs. 3 and 5, the synchronization stability keeps improving for both continuous-time and discrete-time systems as the temporal activity A is increased (to values significantly larger than 0).

Reviewer #3: 4. *The case in which the switching is too slow (in continuous time) or too fast (in discrete time) are dismissed too quickly. There are papers on slow switching that also would need to be referenced. In the continuous time case, it is stated that the network would desynchronize well before any meaningful change occurs, which I think is a reference to a nonlinear effect (but a fair comparison would need to be carried in the linear regime.) For discrete time systems it is stated that there is no fast switching – to which I would object that it is contingent on the specific form of the discrete time system.*

Response: We did not dismiss the slow-switching limit (in continuous time) too quickly. On the contrary, an entire page is dedicated to the discussion on the slow-switching limit (Lines 188-215). A fair comparison among different switching rates ω , in terms of their linear stability, is already presented in Fig. 3b.

As for the fast-switching limit in discrete time, perhaps the referee was thinking about certain discrete approximations of continuous-time systems (e.g., ODE integrators). We have now added the qualifier “usually” in that statement on Line 282. However, note that, unlike in continuous-time systems, the discrete nature of the dynamics precludes the use of the averaging techniques essential for connecting fast-switching networks with their time-averaged counterparts. Thus, one cannot map a temporal network to its time-averaged counterpart in discrete-time systems even when the network topology changes much more rapidly than the node dynamics. These points have been further clarified in the manuscript.

Reviewer #3: 5. *Finally I am confused by the main conclusion that temporal networks can synchronize even when stable synchrony is “impossible in any static network under the given budget”. The actual result is that the temporal network can synchronize if it spends some time in the region in which the Lyapunov exponent is negative (to the right of the first zero in fig. 1) and some time in the region in which the Lyapunov exponent is positive (to the left of the first zero in fig. 1.) Thus the condition for the temporal network to synchronize is that at a certain time it must be “equal” to a static network that can synchronize. Or does that individual instance not satisfy the budget? I find the whole story of the budget, without a proper mathematical definition, very confusing. The word budget is also used in the introduction to indicate the control energy, which is a difference concept.*

Response: Perhaps the referee accidentally mixed the stability of perturbation modes with the stability of the full network. In order for synchronization in a network to be stable, all $n-1$ of its transverse perturbation modes must be stable. In the temporal networks designed by us, although different perturbation modes are unstable at different time instants, the full network is nevertheless stable when evolved over time. In contrast,

any snapshot of the temporal network is unstable, because there are unstable transverse perturbation modes at any given time instant (even when the snapshot is an optimal static network). Thus, our main conclusion that “temporal networks can synchronize even when stable synchrony is impossible in any static network under the given budget” is valid the way it is stated.

Also, just to be clear, our temporal networks satisfy the coupling budget for all time t , which is a key constraint we built into our model and is part of what makes our work interesting. As for the use of the word “budget”, we believe we are using it consistently to refer to the coupling strength in a network (except the broad-picture introduction in the first paragraph of the paper). Further mathematical details on the “coupling budget” are also given on Lines 74-82.

Reviewer #3: *To conclude, my recommendation is to resubmit this work to a more technical journal by making sure to include clear references to relevant previous work on temporal networks with commuting Laplacians and to emphasize the differences of the current work from this previous fundamental work.*

Just to be clear: the matter is not just that a comparison with previous work is missing, it is that a comparison with a key reference is missing. After properly placing the paper in the context of the existing literature, it becomes apparent that the main contribution of the paper is incremental and much more limited than what one would conclude otherwise.

Response: We thank the referee again for the detailed comments, which helped us to improve the paper. We hope that after these revisions and clarifications, the referee will find the revised manuscript now suitable for publication in Nature Communications.

Reviewers' Comments:

Reviewer #2:

Remarks to the Author:

The authors have addressed the comments of the referees. The paper can be accepted for publication in Nature Communications.

Reviewer #3:

Remarks to the Author:

I have read the revised version of the paper and I have found that the authors have failed to address my main criticism that the paper had a large overlap with previous work. In particular:

1) The authors have failed to acknowledge that also their conclusion is exactly the same as that of [18], namely: "that synchronization in a dynamical network can be achieved even in the case in which each individual commutative graphs does not give rise to synchronized behavior" -- see abstract of [18]. It should be noted that [18] also had already pointed out how the results therein differed from the fast switching regime, see this sentence from [18]: ". Finally, we highlight that the present case is by far different from the fast switching procedure described in Ref. ..."

2) In my original review I had pointed out that there are papers focusing on slow switching that also should be cited. From what I can see, the authors have not added any reference to the slow switching regime. One relevant such paper is the following:

Synchronization in slowly switching networks of coupled oscillators
Jie Zhou, Yong Zou, Shuguang Guan, Zonghua Liu & S. Boccaletti
Scientific Reports volume 6, Article number: 35979 (2016)

I copy here from the abstract of this paper:

"Comparison between fast- and slow-switching networks allows elucidating that slow-switching processes prompt synchronization in the cases where the Master Stability Function is concave, whereas fast-switching schemes facilitate synchronization for convex curves."

Also from the discussion of the same paper:

"Specifically, when the MSF is concave (convex) in the domain the argument span, the network supports a synchronous dynamic under a fast (slow) switching, the same network also behaves synchronously under slow (fast) switching of the same structures. This result further exploits the information embedded in the MSF, from the number and positions of the null points to the shape of it."

So this reference already had provided insight "that the effectiveness of introducing time-varying coupling depends critically on the curvature of the master stability function".

It should be noted that this reference is not about "commutative graphs" and so it pertains also to the case of non-commutative temporal networks of the new figure 6.

Once again, in the absence of a clear comparison with previous work in the literature, it is impossible to fairly assess the degree of innovation of the paper. From what I can see, the degree of innovation is low.

Responses to Specific Comments of Reviewer #3

Reviewer #3: *I have read the revised version of the paper and I have found that the authors have failed to address my main criticism that the paper had a large overlap with previous work. In particular:*

1) The authors have failed to acknowledge that also their conclusion is exactly the same as that of [18], namely: "that synchronization in a dynamical network can be achieved even in the case in which each individual commutative graphs does not give rise to synchronized behavior" -- see abstract of [18]. It should be noted that [18] also had already pointed out how the results therein differed from the fast switching regime, see this sentence from [18]: ". Finally, we highlight that the present case is by far different from the fast switching procedure described in Ref. ..."

Response: The reviewer stated that “The authors have failed to acknowledge that also their conclusion is exactly the same as that of [18], namely: ‘that synchronization in a dynamical network can be achieved even in the case in which each individual commutative graphs does not give rise to synchronized behavior’ -- see abstract of [18].” However, as we explained in our last response, this is not what our paper is about. Instead, our key message is that there are temporal networks that synchronize better than *all* static networks when the total coupling budget is limited.

When put side by side, it should be clear that these two statements are very different. There is a significant gap between showing that temporal networks can synchronize better than their time-frozen counterparts versus constructing temporal networks that outperform all possible static networks. A trivial example for the former is a temporal network that switches between many disconnected network configurations, whose time-averaged counterpart becomes connected and synchronizable already in the fast-switching limit.

The reviewer is right that the technique in Ref. [18] can in principle be applied to networks beyond the fast-switching limit, which we now further emphasize in the introduction (Line 63).

However, Ref. [18] does not show that non-fast-switching networks can synchronize better than fast-switching ones. This was only demonstrated explicitly later, for example, in the following papers [Chen, L., Qiu, C. & Huang, H. Phys. Rev. E 79, 045101 (2009); Jeter, R. & Belykh, I. IEEE Trans. Circuits Syst. I, Reg. Papers 62, 1260–1269 (2015); Golovneva, O., Jeter, R., Belykh, I. & Porfiri, M. Physica D 340, 1–13 (2017).]. These studies were already acknowledged in the introduction, and we never claimed novelty over the finding that some temporal networks synchronize only for intermediate switching rate. (We do, however, provide new insights into this intriguing phenomenon and further elucidate the effect of network switching rate on synchronization.)

In summary, the insights presented in Ref. [18], although important, in no way compromise the novelty of our work.

Reviewer #3: 2) *In my original review I had pointed out that there are papers focusing on slow switching that also should be cited. From what I can see, the authors have not added any reference to the slow switching regime. One relevant such paper is the following:*

*Synchronization in slowly switching networks of coupled oscillators
Jie Zhou, Yong Zou, Shuguang Guan, Zonghua Liu & S. Boccaletti
Scientific Reports volume 6, Article number: 35979 (2016)*

I copy here from the abstract of this paper:

"Comparison between fast- and slow-switching networks allows elucidating that slow-switching processes prompt synchronization in the cases where the Master Stability Function is concave, whereas fast-switching schemes facilitate synchronization for convex curves."

Also from the discussion of the same paper:

"Specifically, when the MSF is concave (convex) in the domain the argument span, the network supports a synchronous dynamic under a fast (slow) switching, the same network also behaves synchronously under slow (fast) switching of the same structures. This result further exploits the information embedded in the MSF, from the number and positions of the null points to the shape of it."

So this reference already had provided insight "that the effectiveness of introducing time-varying coupling depends critically on the curvature of the master stability function". It should be noted that this reference is not about "commutative graphs" and so it pertains also to the case of non-commutative temporal networks of the new figure 6.

Once again, in the absence of a clear comparison with previous work in the literature, it is impossible to fairly assess the degree of innovation of the paper. From what I can see, the degree of innovation is low.

Response: We thank the referee for providing this new reference, which is a relevant work that we were previously unaware of. In Zhou et al. [Sci. Rep. 6, 35979 (2016)], the authors study synchronization in slow-switching networks and show that the shape of a master stability function determines whether slow-switching or fast-switching networks synchronize better. This reference has now been cited in the introduction (Line 66) as well as where we discuss the slow-switching limit and the curvature of the master stability function (Lines 192, 210 and 211).

The insights presented in Zhou et al. are important and indeed touch on the role of a master stability function's curvature on synchronization. Below, we would like to discuss some key differences between our results and the results presented in Zhou et al.

First, in Zhou et al., the effect of a master stability function's curvature is discussed exclusively for networks with a fixed topology (only the overall coupling strength is allowed to evolve over time). In our paper, the critical role played by a master stability function's curvature is established for all commutative temporal networks. In particular, the network topology is allowed to be time dependent in our case.

Second, Zhou et al. established a master stability function's curvature as an indicator of whether slow-switching or fast-switching networks synchronize better, whereas here we demonstrate that a master stability function's curvature at its first zero reflects whether temporal networks can outperform optimal static networks, especially when the total coupling budget is limited.

Third, for coupled one-dimensional maps, we establish the role of a master stability function's curvature for *arbitrary* switching rate, as opposed to only for the slow-switching limit. We also show that, for coupled one-dimensional maps, the curvature of the master stability function is always negative. As a consequence, synchronization in these systems can be universally improved by the temporal networks designed in our study.

Fourth, and most importantly, our main conclusion that “temporal networks can synchronize even when stable synchrony is impossible in *any* static network under the given budget” is not presented in Zhou et al., which neither considers the effect of resource constraints nor demonstrates the fundamental synchronization advantage of temporal networks compared to all possible static networks.

Finally, we want to thank the reviewer again for pointing out this important reference, which helped to enrich our discussion of the literature.